# Effect of Phosphogypsum Based Filler on the Performance of Asphalt Mortar and Mixture

**DOI:** 10.3390/ma16062486

**Published:** 2023-03-21

**Authors:** Jiuming Wan, Tao Han, Kaifei Li, Suxun Shu, Xiaodi Hu, Wenxia Gan, Zongwu Chen

**Affiliations:** 1School of Civil Engineering and Architecture, Wuhan Institute of Technology, Wuhan 430205, China; 2Key Laboratory of Road Structure and Material of Ministry of Transport (Changsha), Changsha University of Science & Technology, Changsha 410114, China; 3Local Joint Engineering Laboratory of Traffic Civil Engineering Materials, Chongqing Jiaotong University, Chongqing 400074, China

**Keywords:** phosphogypsum, steel slag powder, asphalt mortar, asphalt mixture, overall desirability

## Abstract

This study introduced phosphogypsum coupled with steel slag powder to prepare the phosphogypsum based filler (PF) for asphalt mixture. Penetration, penetration index, softening point, ductility, equivalent softening point, moisture stability of asphalt mortars with different steel slag powder content, filler-asphalt ratio, and PF content were studied. Mechanical properties of PF based asphalt mortar (P-AM) were then analyzed to determine the optimum steel slag content in PF. Overall desirability method was used to determine the optimum replacement ratio of PF content in limestone filler. Rheological properties of P-AM were also analyzed through dynamic shear rheometer. Volumetric performance, high-temperature performance, low-temperature performance, and moisture stability tests were carried out on PF based AC-20 asphalt mixture. Results showed that P-AM presented the optimum performance when the content of steel slag powder was 23% by mass of phosphogypsum. Fatigue and rutting factor of asphalt mortar were enhanced by PF. The optimum PF content in replacing limestone filler was 75% through overall desirability evaluation. PF developed the high-temperature performance and moisture stability of asphalt mixture. Additionally, volumetric and low-temperature performance were not significantly affected by PF. It is suggested that using PF which is based on phosphogypsum as a filler of asphalt mixture to partially replace traditional limestone filler was adequate.

## 1. Introduction

Construction of asphalt pavement usually consumes a large amount of asphalt, aggregate and filler [1]. Among the natural resources, aggregate constitutes the dominant part of the asphalt mixture [2]. However, great consumption of natural minerals for this purpose resulted in environmental and resource pressure. Therefore, solid wastes have been increasingly used to replace the aggregate and filler in order to reduce consumption of natural mineral resources. Phosphogypsum is a solid waste which is produced from phosphoric chemical industrial processes [3]. Its global annual production is about 280 million tons, which also leads to environmental pollution [4,5,6,7,8]. Therefore, consuming and recycling phosphogypsum as much as possible can benefit environment protection.

It has typically been used as a filler of pavement subgrade and to prepared modified asphalt [9,10,11,12]. Shen et al. [13] prepared phosphogypsum-steel slag powder-flyash as a solidified material of pavement subgrade. Results showed that its early strength and long-term strength were higher than those of cement stabilized granular materials. Rakesh et al. [14] also found the solidified material, which contained fly ash and 8% lime and 2% phosphogypsum, showed adequate unconfined compressive strength, split strength, slake durability criteria and California bearing ratio.

On the other hand, Amrani et al. [15] used 5 wt% phosphogypsum as modifier to prepare modified asphalt. It was found that the modified asphalt showed higher improvements in stiffness and deformation resistance than the values found with fly ash and phosphate sludge wastes. Cuadri et al. [16] used phosphogypsum coupled with sulfuric acid to prepare modified foamed asphalt. Results showed that foamed asphalt based on phosphogypsum has higher rutting resistance and foaming properties compared with the natural gypsum.

However, using phosphogypsum as filler of asphalt mixture has not been systematically studied. Additionally, how to alleviate its negative effect on moisture stability of asphalt material also requires further investigation, since this has hindered its application in road engineering [17]. Alkaline materials should neutralize acidic substances in phosphogypsum, which could improve its water stability and asphalt adhesion. Steel slag is a by-production of steel manufacture, and it has alkalinity and good mechanical properties [18,19,20]. Steel slag based mixtures showed adequate permanent deformation and durable performance [21]. In addition, the asphalt mixture with steel slag powder fillers showed better resistance to moisture damage, and better low-temperature crack resistance, than asphalt mixtures with limestone filler [22]. The steel slag powder is presumed to neutralize acidic substances in the phosphogypsum, thereby enhancing the moisture stability of asphalt mixtures in which phosphogypsum is employed as a filler. 

The concept of overall desirability has been used to integrate optimization based on multiple objectives into a single objective [23]. This method facilitates the integration of indicators with different data ranges into a desirability measure, and can be used to calculated the composition for achieving the optimum performance [24]. Different performance values have been combined into overall desirability to determine the optimum content, with desirability values varying from 0 to 1. Hence, this method has been used to investigate the optimum PF content in this study.

The study aimed to develop a phosphogypsum based filler (PF) which contained steel slag powder, in order to put this waste product to good use. Firstly, the optimum dosage of steel slag powder as modifier was determined through analysis and characterization of PF based asphalt mortar (P-AM). Secondly, the optimized PF was used to partly replace limestone filler in asphalt mortar. Asphalt mortars containing mixed fillers of PF and limestone filler (PL-AM) at different filler-asphalt ratios were prepared and tested. Thirdly, penetration, softening point, ductility and penetration index were used to determine the PF content in replacing traditional limestone filler based on overall desirability. Finally, asphalt mixtures that employed the mixed filler containing PF were fabricated and tested to verify the feasibility of using phosphogypsum as filler after PF composition and content in the mixed filler were determined. The results of this study provide an original approach for consuming the phosphogypsum in asphalt mixture. This approach is positive for reducing phosphogypsum and steel slag, which is beneficial for environmental protection.

## 2. Materials and Methods

### 2.1. Raw Material

#### 2.1.1. Phosphogypsum and Steel Slag Powder

Figure 1 show the appearance of phosphogypsum and steel slag powder. Table 1 shows the main chemical components of phosphogypsum and steel slag powder; the elements are given in form of their oxides. According to XRF analysis, the main component of phosphogypsum is CaSO_4_·2H_2_O and its density is 2.371 g/cm^3^. The physically adsorbed moisture of phosphogypsum will be removed at 100 °C heating. Studies reported that phosphogypsum will then turn into hemihydrate phosphogypsum (CaSO_4_·0.5H_2_O) when the temperature reaches about 130 °C. Hemihydrate phosphogypsum will turn into anhydrous hard phosphogypsum (CaSO_4_) when the temperature exceeds 1200 °C. 

Through XRF analysis, the main component of steel slag is Al_2_O_3_, and its density is 3.498 g/cm^3^. The steel slag powder had been exposed in the air for over 1 year. It was found that the free CaO (f-CaO) of steel slag powder was 2.1%, which was below the upper limit of f-CaO. This indicated that steel slag powder will not lead to volume instability.

#### 2.1.2. Asphalt

Table 2 shows the neat asphalt with penetration range of 60–80 used in this study. Mechanical properties of the asphalt are characterized according to standard testing specification for asphalt and mixture testing (JTG E20-2011, Beijing, in Chinese) and technical specification for construction of highway asphalt pavements (JTG F40-2004, Beijing, in Chinese).

#### 2.1.3. Aggregate and Filler

Table 3 shows the properties of the aggregate and filler, which were tested in accordance with the standards for aggregate testing of highway engineering (JTG E42-2005, Beijing, in Chinese). The results indicated that limestone filler can meet the specification requirements.

### 2.2. Experimental Methods

Figure 2 illustrates the outline of this study. Firstly, this study introduces the preparation of PF. The content of steel slag powder as modifier of PF was set as 0%, 20%, 40%, 60%, 80% and 100%, respectively. The optimum composition of steel slag powder and phosphogypsum was then investigated through characterization of PF based asphalt mortar (P-AM). Secondly, the optimized PF was used to partly replace limestone powder, which formed mixed filler containing PF and limestone powder. PF-L-AM (PL-AM) with filler-asphalt mass ratios of 0.8, 1.0 and 1.2 were prepared. Optimum PF content of the mixed filler was calculated by overall desirability based on the mechanical properties of PL-AM. How filler-asphalt ratio affect PL-AM’s rheological properties was also considered. Finally, asphalt mixtures that employed the mixed filler containing PF were fabricated and tested. The feasibility of using phosphogypsum as filler was then verified after the PF composition and content in mixed filler were determined.

#### 2.2.1. Preparation of PF

This study employed steel slag powder as the modifier in PF since phosphogypsum was acidic and highly hydrophilic, which could lead to poor moisture stability when using phosphogypsum as filler of asphalt mixture alone. Steel slag powder is generally alkaline which should neutralize the acidity of phosphogypsum in a certain degree, benefiting the adhesion between asphalt and aggregate. Phosphogypsum and steel slag powder were first crushed and screened. Both had a particle size less than 0.075 mm.

Both phosphogypsum and steel slag powder were first heated at 135 °C for 5 h. The phosphogypsum (CaSO_4_·2H_2_O) was thus turned into hemihydrate phosphogypsum (CaSO_4_·0.5H_2_O) as pretreatment before preparing PF. The dosage of steel slag powder in PF was designed as 0%, 20%, 40%, 60%, 80% and 100%, respectively, based on preliminary tests. Therefore, 6 kinds of PF were prepared. The optimum content of steel slag powder in PF should be determined according to the mechanical characterization of P-AM.

#### 2.2.2. Preparation of Asphalt Mortar

A high-speed mixing device was used to prepare asphalt mortar, and the mixing temperature was 150 °C. The mixing time was 45 min to ensure the uniformity of materials, and during this time, the rotational speed was varied. The initial rotational speed and time were set as 800 r/min for 15 min after PF or limestone filler were put in with the asphalt. Then, the rotational speed was increased to 1800 r/min for another 15 min. The rotational speed was increased to 3600 r/min for the final 15 min.

PFs of different composition were used to prepare P-AM; their volume was adjusted with that of the limestone filler to avoid the effect of volume difference. The filler-asphalt mass ratios of P-AM were 1:1. Additionally, limestone filler was used to prepare a control group. Thus, by replacing limestone filler at different proportions in equal volume, six kinds of P-AM were prepared to investigate the optimum dosage of steel slag powder based on their mechanical properties.

The optimum PF content in replacing limestone filler was then investigated. The PF content in the mixed filler was 0%, 25%, 50%, 75% and 100%. PL-AM was thus introduced. PL-AM’s filler-asphalt mass ratios were 0.8, 1.0 and 1.2, respectively. The total volume of the mixed filler containing PF and limestone filler was unchanged when preparing PL-AM. Through characterization and analysis of properties, the optimum replacing proportion of PF for limestone filler can be ascertained. Limestone filler asphalt mortar (L-AM) was also prepared as control group.

#### 2.2.3. Experiments Using Asphalt Mortar

##### Determination of P-AM Composition

The influence of steel slag powder content on the penetration, softening point and ductility of P-AM was firstly analyzed according to (JTG E20-2011). The results were then analyzed to figure out the optimum content of steel slag powder. 

In addition, it is necessary to characterize the effect of PF on adhesion between asphalt and limestone, since phosphogypsum is acidic and shows poor moisture stability. How steel slag powder affects the moisture stability of asphalt mixture can hence be investigated. A boiling test investigating the adhesion between P-AM and aggregate was introduced. Limestone aggregate with the size of 26.5 to 31.5 mm was selected. Aggregates were heated at 150 °C and then put in P-AM at 135 °C for 10 s so that they could be fully covered with P-AM. They were then put in boiling water for 30 min after cooling for 24 h at 25 °C. Afterwards, the boiled aggregate was put in a drying box at 80 °C for 5 h to remove moisture. Finally, the mass loss ratio of P-AM on aggregate can be described as Equation (1) [25].
(1)$$\mathrm{MLR}=\frac{m_{1}-m_{2}}{m_{1}-m_{0}}\times 100\%$$
where: MLR = mass loss ratio of P-AM on aggregate, (%);

$m_{0}$ = mass of original aggregate, (g);

$m_{1}$ = mass of aggregate covered by P-AM;

$m_{2}$ = mass of the boiled aggregate.

##### Determination of PL-AM

The penetration, ductility and softening point of different PL-AM mixes were characterized. The penetration index (PI) and equivalent softening point T_800_ were also included to indicate their mechanical properties. Equation (2) is a unary linear equation which was used to fit the functional relationship between penetration and temperature. Equations (3) and (4) were used to calculate PI and T_800_ [26]. The equations used are as follows and all indexes are dimensionless:(2)$$l_{g}P=K+A_{l_{g}\mathrm{Pen}}\times T$$
(3)$$\mathrm{PI}=\frac{20-500A_{l_{g}\mathrm{Pen}}}{1+50A_{l_{g}\mathrm{Pen}}}$$
(4)$$T_{800}=\frac{l_{g}800-K}{A_{l_{g}\mathrm{Pen}}}=\frac{2.9031-K}{A_{l_{g}\mathrm{Pen}}}$$
where: $l_{g}P$ = Penetration logarithm at different temperatures;

K = Constant of the linear equation;

$A_{l_{g}\mathrm{Pen}}$ = Slope of the linear equation;

T = Testing temperature of penetration;

PI = Penetration index;

$T_{800}$ = Equivalent softening point.

This study adopted the overall desirability method which can combine comprehensive indicators and express the final effect by overall desirability. Introduction of comprehensive indicators in the overall desirability method facilitates the integration of indicators with different data ranges into desirability data; thus, the optimum value of asphalt mortar considering each factor control range can be calculated. Using the overall desirability method, dimensional indicators such as penetration, softening point, and ductility are standardized and converted into corresponding dimensionless desirability values between 0 and 1 through linear transformation. The geometric mean of desirability for each indicator can be calculated. Consequently, the desirability of overall evaluation can be obtained. The closer the desirability of the general evaluation to “1”, the better the comprehensive performance of the asphalt mortar will be.

Penetration, penetration index (PI), ductility, and softening point were taken as the calculation indexes of the overall desirability method. Firstly, for each performance indicator (γ), the maximum and minimum value obtained in this study were noted. Then, the desirability ($\gamma_{n}^{*})$ was calculated by linear transformation of performance indicators, using Equations (5) and (6). The equations specify the calculation of desirability based on γ whose values positively or negatively determine the performance of asphalt mortar. Finally, the desirabilities of each performance indicator were used to calculate their geometric mean value by Equation (7), which was the overall desirability. The calculation method of overall desirability is shown below [27,28]:(5)$$\gamma_{\max}^{*}=\frac{\gamma -\gamma_{\min}}{\gamma_{\max}-\gamma_{\min}}$$
(6)$$\gamma_{\min}^{*}=\frac{\gamma_{\max}-\gamma }{\gamma_{\max}-\gamma_{\min}}$$
(7)$$\mathrm{OD}={\left(\gamma_{1}^{*}\gamma_{2}^{*}\gamma_{3}^{*}\ldots \gamma_{n}^{*}\right)}^{\frac{1}{n}}$$
where: OD = overall desirability, dimensionless;

$\gamma_{n}^{*}$ = desirability index after linear transformation, dimensionless; 

γ = performance indicator value, dimensionless; 

n = number of performance indicators used in overall desirability, dimensionless;

$\gamma_{\max}$ = maximum value of corresponding performance indicator in this study, dimensionless; 

$\gamma_{\min}$ = minimum value of corresponding performance indicator in this study, dimensionless;

$\gamma_{\max}^{*}$ = desirability based on the γ whose value positively determined performance of asphalt mortar, dimensionless; 

$\gamma_{\min}^{*}$ = desirability based on the γ whose value negatively determined performance of asphalt mortar, dimensionless.

##### Rheological Performance

Rheological performance of PL-AM and limestone filler were also characterized by using a dynamic shear rheometer (DSR) after PF’s content and composition were determined. A limestone filler-based asphalt mortar with filler-asphalt mass ratio of 1:1 was prepared as control group. Table 4 shows the setting table of DSR high-temperature scanning parameters.

#### 2.2.4. Pavement Performance of PF Based Asphalt Mixture

Mixed filler containing PF was also used to prepared asphalt mixture to indicate its effect on pavement performance. AC-20 was used as the gradation of asphalt mixture which was specified in JTG E40-2005, and its gradation curve is shown in Figure 3. Limestone filler-based AC-20 asphalt mixture was firstly fabricated. Afterwards, PF based asphalt mixture using the mixed filler containing PF can be prepared. The mass ratio of asphalt to aggregate was 4.25% for both the limestone filler and PF based asphalt mixture. Volumetric performance, high-temperature performance, low temperature performance and moisture stability were tested. Volumetric property refers to void volume (VV) and void in mineral aggregate (VMA).

##### High-Temperature Performance

High-temperature performance was evaluated by testing the asphalt mixture’s Marshall stability and dynamic stability according to JTG F20-2011. The dynamic stability test should be applied at 60 °C. Track plate specimens and a dynamic stability instrument were used. Stability can be calculated by Equation (8) [29].
(8)$$\mathrm{DS}=\frac{(t_{2}-t_{1})\times 42}{d_{2}-d_{1}}\times c_{1}\times c_{2}$$
where: DS = dynamic stability of the asphalt mixture, (cycle/mm);

$t_{1}$, $t_{2}$ = test time, usually 45 min and 60 min;

$d_{1}$, $d_{2}$ = deformation of specimen surface corresponding to the test specimens $t_{1}$ and $t_{2}$, (mm).

$c_{1}$, $c_{2}$ = correction factor of testing machine or specimen, dimensionless.

##### Low-Temperature Flexural Performance 

Low-temperature flexural performance of the asphalt mixtures was evaluated by a three-point bending test. Track plate specimens were cut into beam specimens with a size of 250 mm × 30 mm × 35 mm, according to JTG E20-2011. A Universal Testing Machine (UTM-100) was used for the three-point bending test at −10 °C. The distance between supporting fulcrums was 200 mm. The loading rate of the principal axis was 50 mm/s. Corresponding indictors can be calculated using the following equations [30]:(9)$$R_{B}=\frac{3\times L\times P_{B}}{2\times b\times h^{2}}$$
(10)$$\varepsilon_{B}=\frac{6\times h\times d}{L^{2}}$$
(11)$$S_{B}=\frac{R_{B}}{\varepsilon_{B}}$$
where: $R_{B}$ = flexural tensile strength (MPa);

$\varepsilon_{B}$ = tensile strain (με);

$S_{B}$ = tensile stiffness modulus (MPa);

$P_{B}$ = loading peak (kN);

L = span length of beam (mm);

h = height of midspan section (mm);

b = width of midspan section (mm);

d = midspan deflection at failure (mm).

##### Moisture Stability

It is important to characterize the moisture stability of the asphalt mixtures as phosphogypsum’s high hydrophilicity may lead to poor adhesion between asphalt and aggregate. The immersion Marshall stability ratio (IMS) and freeze-thaw tensile strength ratio (TSR) were used to comprehensively evaluate the moisture stability of asphalt mixture according to (JTG E20-2011). IMS can be calculated by Equation (12) [29].
(12)$$\mathrm{IMS}=\frac{{\mathrm{MSR}}_{1}}{\mathrm{MSR}}\times 100\%$$
where: MSR: the average stability of specimen in moisture at 60 °C for 30 min (kN);

${\mathrm{MSR}}_{1}$: the average stability of specimen in moisture at 60 °C for 48 h (kN); 

IMS: the average residual stability of specimen in moisture.

On the other hand, TSR can be calculated by Equation (13) [29].
(13)$$\mathrm{TSR}=\frac{R_{T2}}{R_{T1}}\times 100$$
where: TSR: the average strength ratio of the freeze-thaw splitting test;

$R_{T1}$: splitting tensile strength of the specimens without freeze-thaw cycle (the unconditional); 

$R_{T2}$: splitting tensile strength of specimens after freeze-thaw cycle (the conditional).

## 3. Results and Discussions

### 3.1. Determination of PF Composition

#### 3.1.1. Physical Properties of P-AM

Table 5 and Table 6 show the results of penetration, ductility and softening point tests for P-AM. The penetration of P-AM showed the lowest value when 20% steel slag powder was added, at which the highest consistence as well as high-temperature performance of P-AM was achieved. The softening point of P-AM was higher than that of L-AM when 0–60% steel slag powder was mixed. P-AM with 20% steel slag powder showed the highest softening point, suggesting the maximum high-temperature performance. These findings indicate that an appropriate mixture of phosphogypsum and steel slag powder results in an improved softening point. However, the ductility values of P-AM were significantly lower than those of limestone asphalt mortar. The content of steel slag powder showed no statistically significant effect on the ductility of P-AM. It was evident that an excessive content of steel slag powder will result in poor physical properties. Furthermore, it was speculated that there could be a coupling effect of phosphogypsum and steel slag powder which could determine the properties of asphalt mortar. Consequently, the optimum content of steel slag powder should be determined after comprehensive consideration of P-AM’s physical properties. A functional curve fitting the data of penetration and softening point of P-AM was used to assess the effects of steel slag powder content intuitively, as shown in Figure 4.

Functional curves of the data points were then fitted to show how the content of steel slag powder affected the properties of P-AM. The equation of the fitting curve for the softening point was found to be:y = 22.569x^3^ − 40.327x^2^ + 16.27x + 55.048

The determination coefficient (R²) was 0.9904, which indicated that the fitting of the softening point was reliable enough. On the other hand, the equation of the fitting curve for penetration was found to be:y = −73.958x^3^ + 142.05x^2^ − 59.917x + 53.843

Its R^2^ was 0.9572, which also suggested that the fitting of penetration was adequate. It was found that the highest softening point and lowest penetration occurred when the content of steel slag powder was 23%. The highest high-temperature performance, stiffness and plasticity can be achieved at this content. Since the content of steel slag powder showed no significant effect on the ductility of P-AM, the optimum content of steel slag powder was determined as 23% of PF volume considering its contribution to the softening point and penetration.

#### 3.1.2. Adhesion Characterization

Images of aggregates covered with P-AM film after the boiling test are shown in Figure 5. It appears that the P-AM film on aggregate with 0% steel slag powder seriously peeled off. However, P-AM films contained steel slag powder showed no obvious spalling after the boiling test. This result indicates that adding steel slag powder to PF can enhance adhesion between the aggregate and the asphalt binder. P-AM without steel slag powder showed poor adhesion with aggregate due to acidity of phosphogypsum. In contrast, alkaline steel slag powder can neutralize acidity of phosphogypsum to a certain degree, so that adhesion between the aggregate and the asphalt binder can be developed. This result further indicates that steel slag was positive for the moisture resistance of the asphalt mixture which is correlated to adhesion between asphalt and aggregate. However, how the content of steel slag powder affected adhesion was hard to conclude from appearances since the spalling was not easily quantifiable (Figure 5).

The mass loss percentage of P-AM is shown in Figure 6, and enables quantitive characterization of the adhesion of P-AM. It was found that P-AM without steel slag powder was nearly fully removed by boiling water, showing that using pure phosphogypsum as filler for asphalt mixture was vulnerable to moisture damage. Mass loss percentages of P-AM were significantly reduced for steel slag powder contents over 20% of PF, proving that steel slag powder can effectively enhance adhesion between asphalt and aggregate. The mass loss percentage of P-AM did not show monotonic reduction as more steel slag powder was introduced, however. It is believed that the enhancement effect of steel slag powder on adhesion is limitative, so that continuously increasing the content of steel slag powder cannot further develop adhesion. The content of steel slag powder within PF was hence suggested to be more than 20% to achieve adequate adhesion enhancement. Consequently, considering the optimum content of steel slag powder as illustrated in Figure 4 and adhesion characterization as illustrated in Figure 6, the optimum volume percentage of steel slag in PF was determined as 23%.

### 3.2. Effect of PF Content on PL-AM’s Mechanical Properties

#### 3.2.1. Penetration

Figure 7, Figure 8 and Figure 9 show the penetration results of PL-AM with filler-asphalt ratios of 0.8, 1.0 and 1.2. Different PF content and testing temperatures were also included. Testing temperature was positively correlated to penetration value due to the viscoelastic characteristics of asphalt. Penetration value showed first a decreasing and then an increasing tendency along with the increase in PF content, independent of temperature. PL-AM presented the lowest penetration results when 75% limestone filler was replaced by an identical volume of PF, regardless of the filler-asphalt ratio and temperature. PL-AM with a higher filler-asphalt ratio shows a lower penetration value since PF is a rigid material. This result suggested that there was a proper composition of filler and PF which achieved the highest stiffness of PL-AM. It was also speculated that there could be a coupling effect of PF and limestone filler so that the effect of PF content on PL-AM’s physical property was not monotonic.

#### 3.2.2. Softening Point

Figure 10 shows PL-AM’s softening point at different PF content with filler-asphalt ratios of 0.8, 1.0 and 1.2. It is evident that increased filler-asphalt ratio leads to a higher softening point owing to the enhancement to the filler. The softening point showed a slightly increasing tendency as PF content was raised from 0% to 75%, and decreased when limestone filler was totally replaced by PF, independent of the filler-asphalt ratio. This shows that optimum PF content in the mixed filler was 75% for achieving the highest softening point, which positively determined the high-temperature performance of the corresponding asphalt mixture. On the other hand, it should be noted that the improvement in the softening point caused by replacing limestone filler with PF was not that significant, considering the fact that softening point difference among different PF content was not over 4 °C.

#### 3.2.3. Ductility

Figure 11 presents the ductility of PL-AM at different filler-asphalt ratios and PF content. It was found that higher filler-asphalt ratio led to lower ductility regardless of PF content. Ductility showed a decreasing tendency as the PF content increased from 0% to 50%, but then developed when PF content was over 75%. The lowest ductility value was found when PF content was 50% regardless of the filler-asphalt ratio, and ductility of PL-AM was the second highest when 75% PF was added. The coupling effect of PF and phosphogypsum probably affected the ductility of PL-AM. It illustrated that replacing limestone filler with PF will negatively affect ductility of mortar, which resulted in lower plasticity. However, this negative effect can be reduced by replacing limestone powder with 75% PF. The results suggested that using PF as filler might negatively affect the low-temperature performance of the corresponding asphalt mixture [31].

#### 3.2.4. PI and T_800_

Penetration at 15 °C, 25 °C and 30 °C was used to calculate the penetration index (PI) and equivalent softening point (T_800_) of asphalt mortar. The logarithmic values of penetration (log P) and temperature (T) were used to calculate PI according to the fitting curves of Equation (2). The linear regression correlation coefficient R^2^ in Equation (2) must not be less than 0.997. Equation (3) shows the calculation of PI, which was negatively correlated to asphalt mortar’s temperature sensitivity. T_800_ can be calculated based on Equation (4) and illustrated the high-temperature stability of asphalt mortar. Table 7 shows the results for penetration index PI, T_800_, regression equation and R^2^ according to their dependence on PF content and filler-asphalt ratio; PF content 0% is the limestone asphalt mortar that was used as control group. Higher values of PI and T_800_ suggested a stronger ability of asphalt mortar in resisting high-temperature deformation. 

Both PI and T_800_ values of PL-AM were larger than that of L-AM regardless of filler-asphalt ratio. Thus, PF reduced the temperature sensitivity of PL-AM. On the other hand, either PI and T_800_ first increased and then decreased with increase of PF content. Additionally, PI and T_800_ also showed an increasing trend as the filler-asphalt ratio was increased. PL-AM with 75% PF achieved the highest PI and T_800_, optimally improving the high-temperature performance. It is speculated that these properties were also related to a coupling effect of PF and phosphogypsum because they did not show a monotonic tendency as PF content was increased. This result was consistent with the penetration and softening point of PL-AM, showing that 75% could be the optimum content of PF in replacing limestone filler.

### 3.3. Determination of PF Content

Table 8 shows the results of the overall desirability calculation. It can be concluded that the total evaluation desirability value of PL-AM had achieved its maximum when PF content was 75%. This proved that its high-temperature performance was optimally enhanced as 75% PF was introduced. PL-AM’s penetration and ductility decreased, while its softening point and PI were the highest at 75% PF, independent of the filler-asphalt ratio. Thus, 75% PF is suggested especially concerning high-temperature performance, consistent with the results for penetration, softening point, PI and T_800_. Therefore, it is suggested that the overall desirability method is a feasible approach to finding the optimum PF content.

### 3.4. Rheological Properties

Figure 12, Figure 13 and Figure 14 shows the results of the DSR high-temperature scanning test for PL-AM with 75% PF at filler-asphalt ratios of 0.8, 1.0 and 1.2, respectively. The complex shear modulus G* of asphalt mortars showed a decreasing trend, while the phase angle showed a rising tendency with increase of temperature. G* of PL-AM was higher than that of L-AM. PL-AM showed higher δ than that of L-AM from 30 to 60 °C, while the contrary result was found from 60 to 80 °C. A higher filler-asphalt ratio resulted in higher G* of PL-AM, while δ was negatively affected by the filler-asphalt ratio from 65 to 80 °C.

PL-AM had a greater rutting factor (G*/sinδ) compared with L-AM when their filler-asphalt ratio was 1.0. This proved that PF was able to increase the hardness of asphalt mortar, so that the ability to resist deformation in high temperatures, namely high-temperature performance, was improved. On the other hand, the fatigue factor (G*sinδ) of PL-AM was also higher than that of L-AM at the same filler-asphalt ratio. Both rutting and fatigue factors were improved by a higher filler-asphalt ratio. Thus, use of a mixed filler containing PF could help to enhance the rutting and fatigue resistance of PL-AM. The corresponding PF based asphalt mixture’s high-temperature and fatigue performance should be higher than a limestone filler based asphalt mixture.

### 3.5. Pavement Performance

#### 3.5.1. Volumetric Performance

P-AM results suggested that the optimum composition of PF was 23% steel slag powder and 77% phosphogypsum as discussed above. PL-AM results showed that the optimum PF content was 75%. Hence, AC-20 asphalt mixtures with the mixed filler containing PF were prepared. A corresponding asphalt mixture without PF was also fabricated to indicate how PF affects pavement performance. Table 9 shows the optimum asphalt-aggregate mass ratio and volumetric performance of the two kinds of asphalt mixtures. It indicates that using PF to partly replace limestone powder as filler of asphalt mixture showed no clear impact on the optimum asphalt-aggregate mass ratio and volumetric performance.

#### 3.5.2. High-Temperature Performance

Figure 15 shows Marshall stability and dynamic stability of the asphalt mixtures. The dynamic stability values of the two kinds of asphalt mixtures both meet the requirement of (JTG F40-2004), which is not less than 800 cycles/mm. The dynamic stability of limestone filler based asphalt mixture was 922 cycles/mm and that of the PF based asphalt mixture was 1265 times/mm, which was improved by 37.2%. In addition, Marshall stabilities of both asphalt mixtures were higher than the requirement of 8 kN. The PF based asphalt mixture showed a higher Marshall stability than that of the limestone filler based asphalt mixture. This showed clearly that PF could improve the high-temperature performance of the asphalt mixture by partly replacing limestone filler. This result was consistent with the softening point test and DSR high-temperature scanning test results, which also suggested that PF can enhance high-temperature performance.

#### 3.5.3. Low-Temperature Flexural Performance

Table 10 shows the test results for low-temperature flexural performance. The flexural-tensile strain of the asphalt mixture with limestone filler was 2249, while the strain for the PF based asphalt mixture was 2162. They can meet the requirement in the JTG F40-2004 specification that flexural-tensile strain should not be less than 2000, indicating that their low-temperature flexural performance was adequate. The flexural tensile stiffness modulus and flexural tensile strength stiffness modulus of the PF based asphalt mixture were higher than those of the limestone filler based asphalt mixture. Thus, using PF as filler would not significantly damage the low-temperature performance of the asphalt mixture.

#### 3.5.4. Moisture Stability

Moisture stability was a key performance criterion since phosphogypsum is acidic and has poor moisture stability. Table 11 and Table 12 present results of IMS and TSR, respectively. IMS results showed that both the PF based asphalt mixture and the limestone filler based asphalt mixture could meet the JTG F40-2004 specification, requiring not less than 80%. The TSR values of both the PF based asphalt mixture and the limestone filler based asphalt mixture were higher than 75%, which meets the specification requirement. On the other hand, the IMS and TSR of the PF based asphalt mixture were higher than the values for the asphalt mixture without PF, proving that using PF was positive for improving the asphalt mixture’s moisture stability. Therefore, it was analyzed that the negative effect of phosphogypsum on moisture stability was offset by the addition of steel slag powder. To conclude, the PF based asphalt mixture showed better high-temperature and moisture stability and adequate low-temperature performance. The results showed that using phosphogypsum based filler containing steel slag powder to partly replace limestone filler was able to develop asphalt mixture’s pavement performance.

## 4. Conclusions

This study focused on recycling phosphogypsum as an ingredient of an asphalt mixture filler. Phosphogypsum and steel slag powder were mixed to fabricate the phosphogypsum based filler (PF). PF based asphalt mortar (P-AM) was prepared. Penetration, softening point, ductility and boiling tests were firstly conducted to determine the optimum content of steel slag powder in PF. PF-limestone based asphalt mortars were prepared and characterized. The overall desirability method was applied to determine the optimum content of PF in replacing limestone filler in the asphalt mixture. Finally, the PF based asphalt mixture was fabricated and tested to verify the feasibility of using phosphogypsum as an asphalt mixture filler. According to the laboratory test results, the following conclusions can be drawn.
(1)PF enhanced asphalt mortar’s softening point and ductility, while penetration was reduced. Steel slag powder clearly improved the adhesion between P-AM and aggregate when its content was over 20%. The highest softening point and lowest penetration occurred when content of steel slag powder was 23%, according to the fitting curve. (2)PL-AM presented the lowest penetration when 75% limestone filler was replaced by PF. Similarly, the highest softening point, penetration index and equivalent softening point were also found when PF content was 75%. The filler-asphalt ratio was positively correlated to softening point, penetration index and equivalent softening point, while it negatively affected penetration and ductility. The overall desirability achieved the maximum value when PF content was 75% based on PL-AM mechanical performance. Consequently, the optimum PF content was determined as 75% in replacing limestone filler.(3)The complex shear modulus of PL-AM was improved by PF. PL-AM showed a higher phase angle from 30 to 60 °C, while the contrary result was found from 60 to 80 °C. PF showed no significant effect on volumetric and low-temperature performance. High-temperature and moisture stability performance were improved by PF. Thus, using phosphogypsum based filler containing steel slag powder to partly replace limestone filler improved the asphalt mixture’s pavement performance.

## Figures and Tables

**Figure 1 materials-16-02486-f001:**
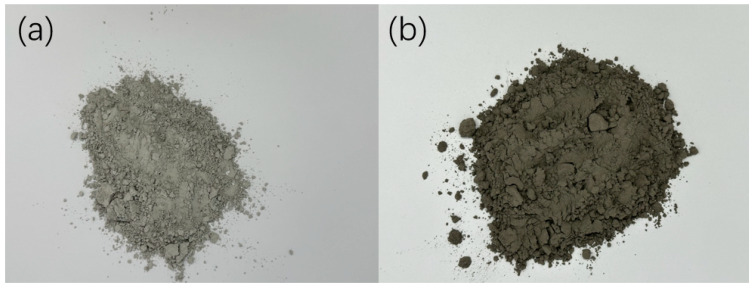
(**a**) Phosphogypsum powder, (**b**) steel slag powder.

**Figure 2 materials-16-02486-f002:**
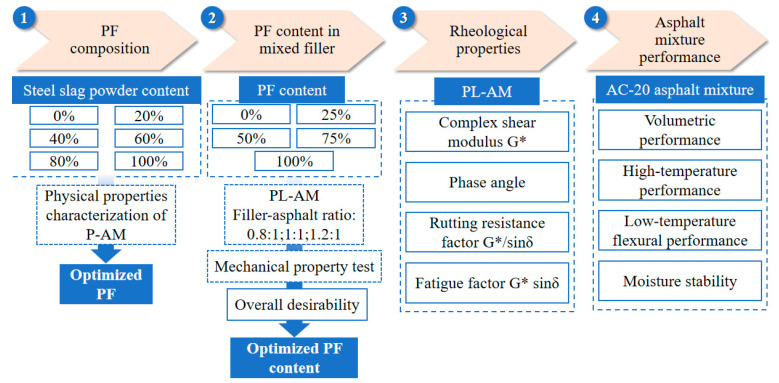
Outline of study.

**Figure 3 materials-16-02486-f003:**
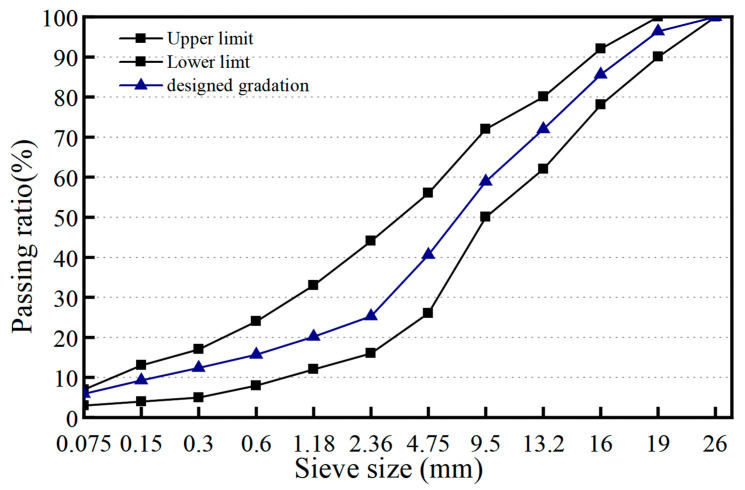
Aggregate gradation curve of AC-20.

**Figure 4 materials-16-02486-f004:**
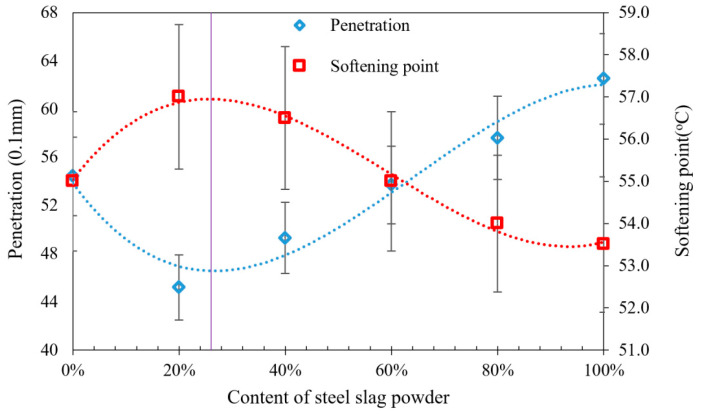
Results of penetration and softening point of P-AM.

**Figure 5 materials-16-02486-f005:**
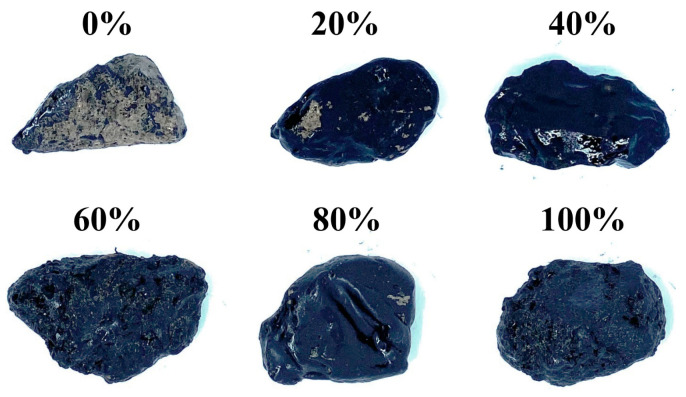
Aggregates with P-AM after boiling test.

**Figure 6 materials-16-02486-f006:**
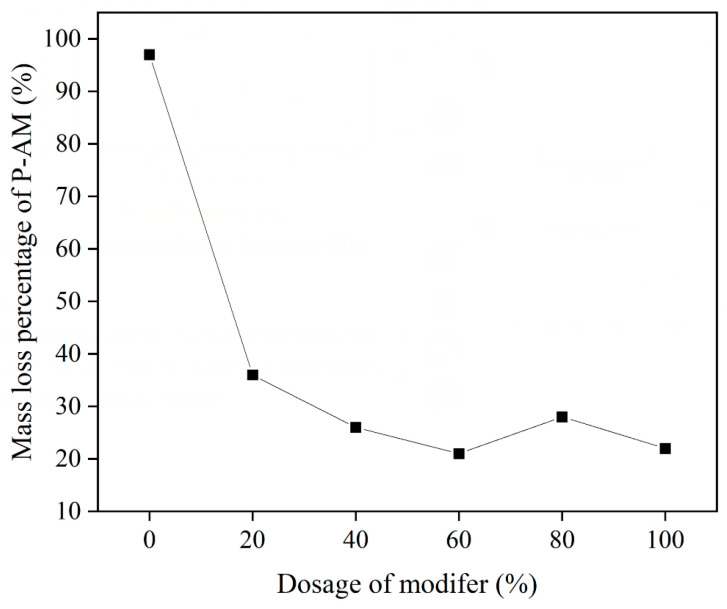
Mass loss percentage of P-AM.

**Figure 7 materials-16-02486-f007:**
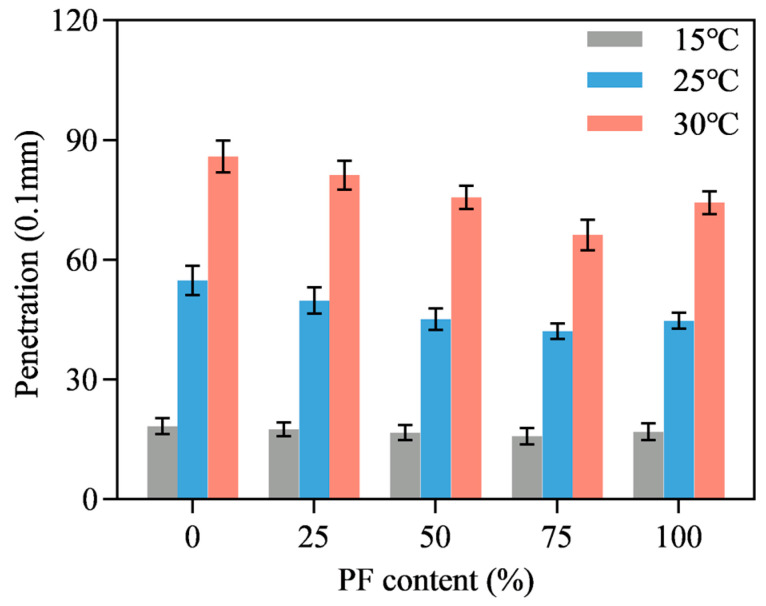
Filler-asphalt ratio of 0.8.

**Figure 8 materials-16-02486-f008:**
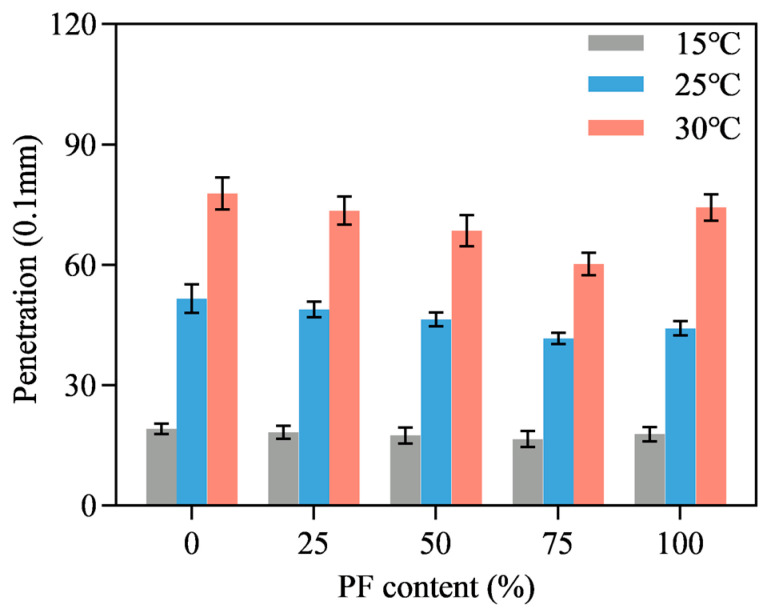
Filler-asphalt ratio of 1.0.

**Figure 9 materials-16-02486-f009:**
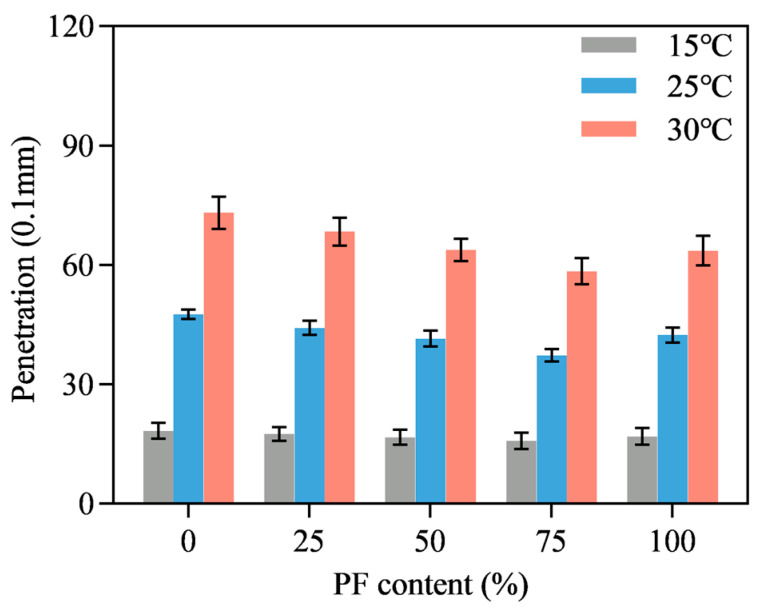
Filler-asphalt ratio of 1.2.

**Figure 10 materials-16-02486-f010:**
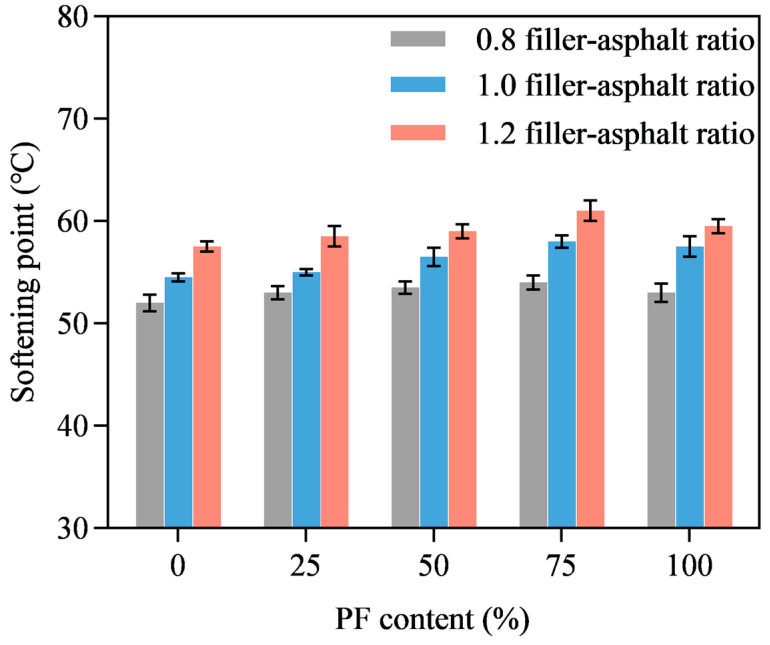
Softening point.

**Figure 11 materials-16-02486-f011:**
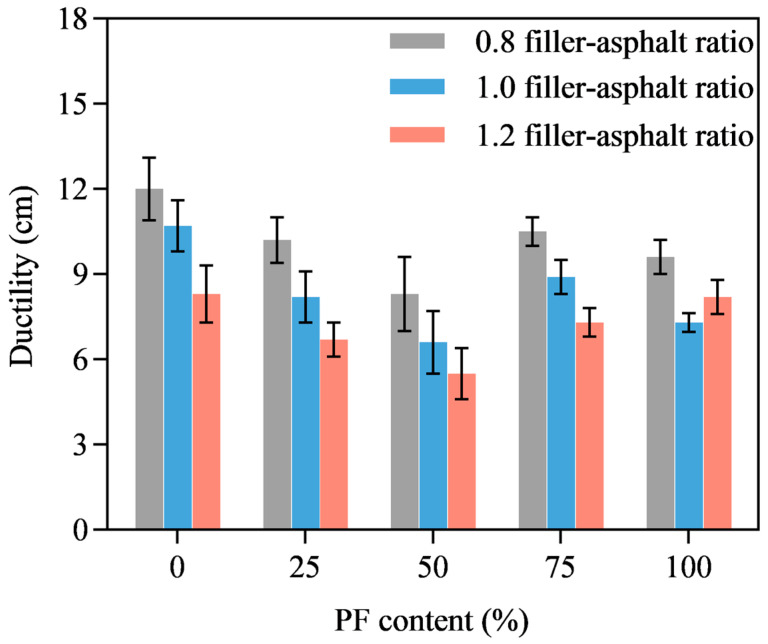
Ductility.

**Figure 12 materials-16-02486-f012:**
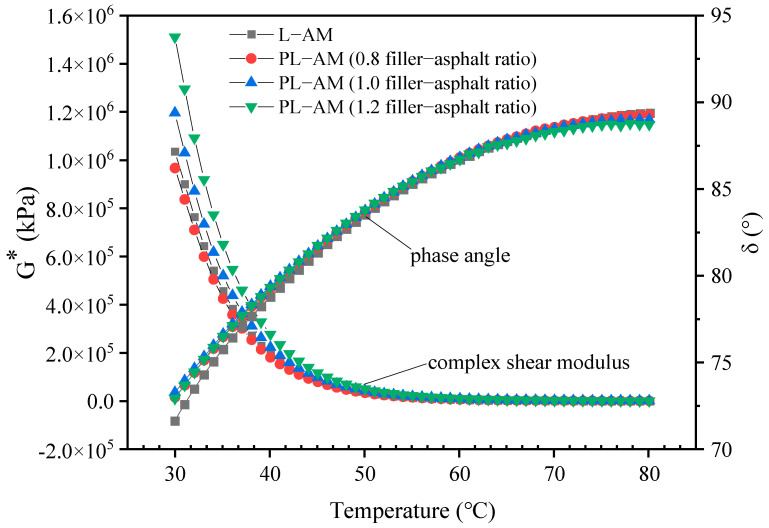
G* and δ of PL-AM and L-AM.

**Figure 13 materials-16-02486-f013:**
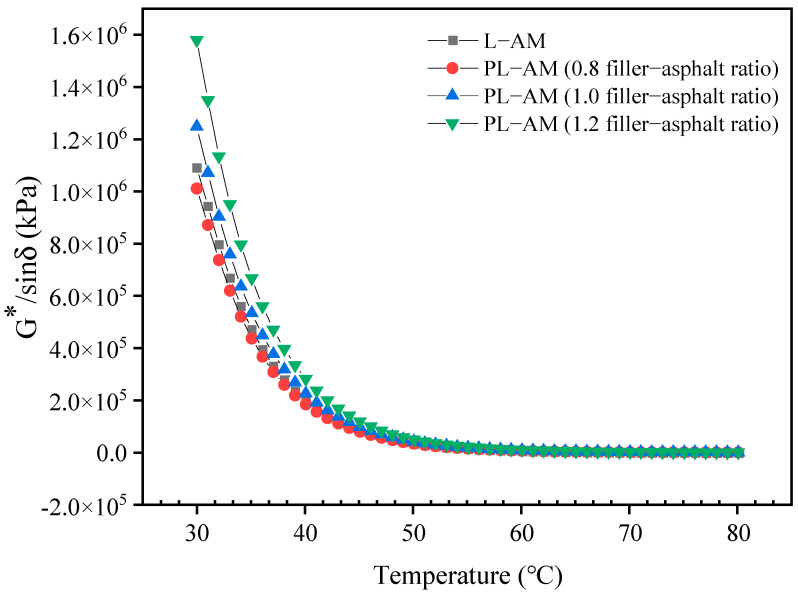
Rutting factor G*/sinδ of PL-AM and L-AM.

**Figure 14 materials-16-02486-f014:**
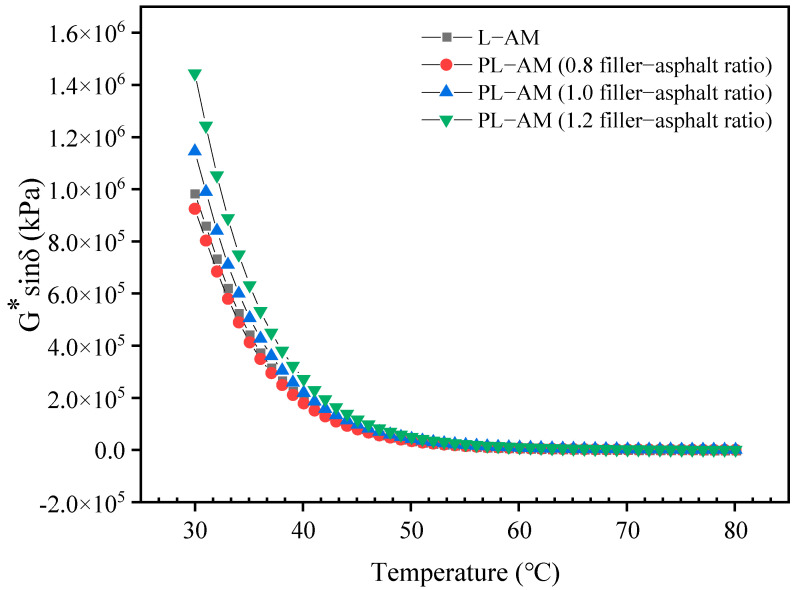
Fatigue factor G*sinδ of PL-AM.

**Figure 15 materials-16-02486-f015:**
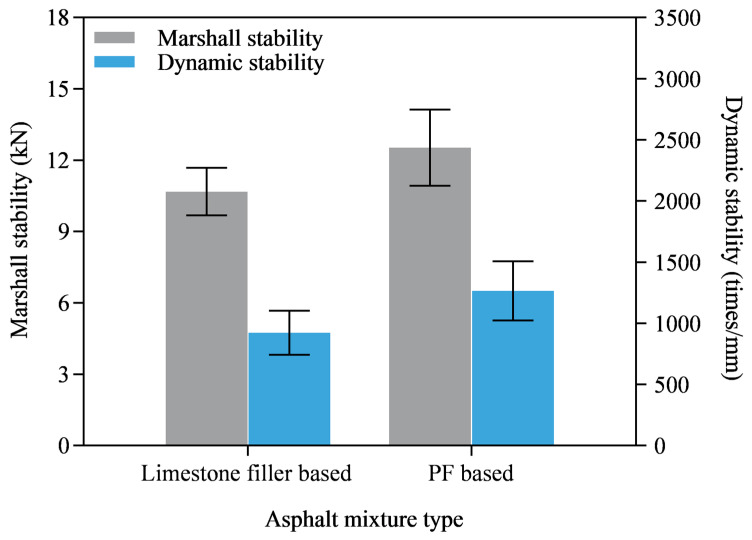
Marshall stability and dynamic stability of asphalt mixture.

**Table 1 materials-16-02486-t001:** The chemical composition of phosphogypsum and steel slag powder.

Composition	SO_3_	Al_2_O_3_	SiO_2_	CaO	P_2_O_5_	Fe_2_O_3_	MgO
Phosphogypsum	44.5%	0.9%	9.5%	31.1%	2.5%	0.7%	/
Steel slag powder	/	22.2%	43.9%	17.8%	/	2.9%	5.7%

**Table 2 materials-16-02486-t002:** Mechanical properties of asphalt.

Technical Index	Test Results	Requirements
Penetration (25 °C, 0.1 mm)	70.7	60–80
Softening point (°C)	49.0	≥46
Ductility (15 °C, cm)	>100	≥100
Viscosity (135 °C, Pa·s)	0.46	/

**Table 3 materials-16-02486-t003:** The basic properties of the aggregate and filler.

Technical Index			Test Results	Requirements
Aggregate	Apparent specific density		2.851	≥2.5
	Crush value (%)		20.7	≤28
	Water absorption (%)		0.8	≤3.0
	Adhesion level		5	5
Filler	Granularity range (%)	<0.6 mm	100	100
		<0.15 mm	91.5	90–100
		<0.075 mm	79	75–100
	Apparent specific density		2.786	/
	Appearance		Agglomerate free of caking	Agglomerate free of caking

**Table 4 materials-16-02486-t004:** DSR high-temperature scanning parameter setting table.

Temperature	Angular Frequency	Rotor Size	Gap Size	Heating Up Speed
30–80 °C	0.1 Hz	25 mm	1 mm	2 °C/120 s

**Table 5 materials-16-02486-t005:** Physical results of P-AM.

Content of Steel Slag Powder	Penetration (25 °C, 0.1 mm)	Ductility (15 °C, cm)	Softening Point (°C)
0%	54.4	3.6	55.0
20%	45.2	4.8	57.0
40%	49.3	4.3	56.5
60%	53.7	3.9	55.0
80%	57.6	4.5	54.0
100%	62.5	4.7	53.5

**Table 6 materials-16-02486-t006:** Results of L-AM.

Penetration (25 °C, 0.1 mm)	Ductility (15 °C, cm)	Softening Point (°C)
51.6	10.7	54.5

**Table 7 materials-16-02486-t007:** Results of PI and T800.

Filler-Asphalt Ratio	PF Content	PI	T_800_	Regression Equation	R^2^
0.8	25%	−0.196	54.0	y = 0.0412x + 0.6705	0.9999
0.8	50%	−0.050	55.5	y = 0.0403x + 0.6605	0.9985
0.8	75%	0.256	58.0	y = 0.0385x + 0.6652	0.9999
0.8	100%	0.067	56.5	y = 0.0396x + 0.6727	0.9985
1.0	25%	−0.099	55.5	y = 0.0406x + 0.6586	0.9980
1.0	50%	0.017	56.5	y = 0.0389x + 0.6406	0.9998
1.0	75%	0.363	59.5	y = 0.0378x + 0.6309	0.9999
1.0	100%	−0.180	55.5	y = 0.0411x + 0.6299	0.9989
1.2	25%	0.067	57.0	y = 0.0396x + 0.6510	0.9998
1.2	50%	0.187	58.0	y = 0.0389x + 0.6406	0.9998
1.2	75%	0.381	60.0	y = 0.0378x + 0.6309	0.9999
1.2	100%	0.239	58.5	y = 0.0386x + 0.6523	0.9991
0.8	0%	−0.338	53.0	y = 0.0421x + 0.6767	0.9993
1.0	0%	−0.164	54.5	y = 0.0410x + 0.6711	0.9980
1.2	0%	−0.050	55.5	y = 0.0403x + 0.6610	0.9993

**Table 8 materials-16-02486-t008:** Overall desirability.

Filler-Asphalt Ratio	PF Content	Penetration (25 °C, 0.1mm)	PI	Ductility (cm)	Softening Point (°C)	OD Value
0.8	0%	54.8	−0.338	12.0	52.0	0
	25%	49.8	−0.196	10.2	53.0	0.268
	50%	45.1	−0.050	8.3	53.5	0.364
	75%	42.1	0.256	10.5	54.0	0.576
	100%	44.7	0.067	9.6	53.0	0.402
1.0	0%	51.6	−0.164	10.7	54.5	0.320
	25%	48.9	−0.099	8.2	55.0	0.358
	50%	46.4	0.017	6.6	56.5	0.380
	75%	41.7	0.363	8.9	58.0	0.716
	100%	44.2	−0.180	7.3	57.5	0.391
1.2	0%	47.6	−0.050	8.3	57.5	0.460
	25%	44.2	0.067	6.7	58.5	0.465
	50%	41.5	0.187	5.5	59.0	0
	75%	37.3	0.381	7.3	61.0	0.730
	100%	42.4	0.239	8.2	59.5	0.671

**Table 9 materials-16-02486-t009:** Optimum asphalt-aggregate mass ratio and volumetric performance.

Volume Parameter	PF Based	Limestone Filler Based
Optimum asphalt-aggregate ratio	4.2%	4.2%
VV	4.12%	4.0%
VMA	13.2%	13.0%
VFA	69.3%	69.0%

**Table 10 materials-16-02486-t010:** Low-temperature flexural performance results.

Types of Asphalt Mixture	Flexural Tensile Strain (με)	Flexural Tensile Strength (MPa)	Flexural Tensile Strength Stiffness Modulus (MPa)
PF based	2162	9.7	4486.59
Limestone filler based	2249	9.3	4135.17

**Table 11 materials-16-02486-t011:** IMS results.

Types of Asphalt Mixture	MSR_1_ (60 °C for 48 h, kN)	MSR (60 °C for 30–40 min, kN)	IMS (%) ≥ 80
PF based	11.11	12.53	88.7
Limestone filler based	8.95	10.68	83.8

**Table 12 materials-16-02486-t012:** TSR results.

Types of Asphalt Mixture	R_T1_ (MPa)	R_T2_ (MPa)	TSR (%) ≥ 75
PF based	0.990	0.797	80.5
Limestone filler based	0.895	0.698	78.0

## Data Availability

Not applicable.

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
