# Peer review of "Effect of Phosphogypsum Based Filler on the Performance of Asphalt Mortar and Mixture"

_materials, 2023, doi:10.3390/ma16062486_

Round 1

Reviewer 1 Report

The introduction fails to identify a gap that this manuscript intends to contribute, even (almost) appropriate literature data are mentioned. Accordingly, please elaborate what is the weakness of previous studies, and why this study is meaningful and necessary.

The abstract is a shortened version of the manuscript and should entice readers to read the full text by highlighting the most important findings/ values. Keeping the abstract concise while covering the important information in an attractive mode requires careful writing and revision. Please shorten/ revise the abstract section accordingly.

Please remove Figure 4, while keeping only the corresponding discussion.

Please reconsider the color used in Figures 9-13 in a more “technical way”, not a PowerPoint presentation.

Please consider merging Tables from Section 2.1.

Please add appropriate literature data foe equations used.

The discussion presented is poor, in terms of discussing its results and comparing them with the bibliography. I suggest reviewing this part more carefully and discuss further. Example: this section should be better presented in order to highlight the most significant and unexpected results, identify correlations, patterns and relationships among the data, speculations, limitations of work and deductive arguments. Please carefully apply.

The Conclusions section should be rewritten/ shorten in order to highlight the following aspects: (i) the problem statement addressed in the paper; (ii) summarize overall findings and (iii) the key takeaways from your paper. Please use relevant values/ numbers.

Finally, I believe that the work is not suitable for publication in this form and requires large additions. If the manuscript will not be considerable improved, I will not recommend its publication.

Author Response

Reviewer #1

1.The introduction fails to identify a gap that this manuscript intends to contribute, even (almost) appropriate literature data are mentioned. Accordingly, please elaborate what is the weakness of previous studies, and why this study is meaningful and necessary.

Response: Thanks for your constructive comments. We do agree with you that this introduction requires large revision. Whole introduction had been rewritten according to your comments. Literature and meaning were also reorganized to imply the necessarily of this study.

2.The abstract is a shortened version of the manuscript and should entice readers to read the full text by highlighting the most important findings/ values. Keeping the abstract concise while covering the important information in an attractive mode requires careful writing and revision. Please shorten/ revise the abstract section accordingly.

 Response: Thanks for your comments. Some redundance in the abstract had been deleted according to your comments and the abstract had been shorten and revised.

3.Please remove Figure 4, while keeping only the corresponding discussion.

Response: Thanks for your comment. We have removed Figure4 and made corrections to the corresponding discussion section.

4.Please reconsider the color used in Figures 9-13 in a more “technical way”, not a PowerPoint presentation.

Response: Thanks for your comment. We have adjusted the color of the Figure 9-13 in this paper according to your comment.

5.Please consider merging Tables from Section 2.1.

Response: Thanks for your comment. According your suggestion, we have merged the tables that can be merged in 2.1. In addition, the introduction section of the table has been amended accordingly.

6.Please add appropriate literature data foe equations used.

Response: Thanks for your comment. We truly believed that it is necessary to add literature date foe equations. Thus, we consulted relevant literature and cite the corresponding literature data.

7.The discussion presented is poor, in terms of discussing its results and comparing them with the bibliography. I suggest reviewing this part more carefully and discuss further. Example: this section should be better presented in order to highlight the most significant and unexpected results, identify correlations, patterns and relationships among the data, speculations, limitations of work and deductive arguments. Please carefully apply.

Response: Thanks for your comment. We do agree with your comments for the discussion part. This part had been reconsidered and some necessary discussion had been added or emphasized. Relationship and speculation between data and corresponding tendency were also discussed.

8.The Conclusions section should be rewritten/ shorten in order to highlight the following aspects: (i) the problem statement addressed in the paper; (ii) summarize overall findings and (iii) the key takeaways from your paper. Please use relevant values/ numbers.

Response: Thanks for your comment. This comment is definitely constructive and helpful, we do agree that conclusion should be revised according to the advised structure and recommendation. Therefore, conclusion had been carefully revised and shorten to highlight the key part of this manuscript.

9.Finally, I believe that the work is not suitable for publication in this form and requires large additions. If the manuscript will not be considerable improved, I will not recommend its publication.

Response: Thanks for your comment. We appreciate that your constructive suggestion had obviously helped us in improving the quality of this manuscript. A lot of revision had been already done refer to all comments, and many required contents was also included. Abstract, introduction, discussion and conclusion were reorganized and rewritten. Quality of this manuscript had been greatly enhanced due to your suggestion. We hope this manuscript can be considered and agreed for publication, after all, this manuscript had been revised in detail.

Thank you again for your rigorous attitude and helpful suggestions in the review.

Reviewer 2 Report

1) What exactly authors mean by "asphalt mortar and mixture" ? Mixture? or matrix?

2) Line 13:that are often : is often

3) As per l 13: it is already used widely in bitumen matrix, then what is the novelty of this work?

4) How the mix composition is arrived? Is it based on DOE or literature or by preliminary tests?

5) Why phospogypsum is mixed with steel slag, what benefits authors expected, and what property of this two materials made authors decide this ?

6) What is the importance of experiment done and explained via fig 4?

7) Line 235-237 is it relative density of soil formula? On what literature basis this calculation is done?

8) Lot of articles content is about explaining the process of experiments, results and discussion is too limited, instead of this authors can simply mention the code used for the test unless the test is defined by authors or used newly which is not covered in a code.

9) Line 297:phogypsum’s high hydrophilicity may lead to poor adhesion between asphalt and aggregate: Then why is this material prescribed if it affects the bonding which in turn will affect strength?

10) Fig 6 interpreting is confusing.

11) Fig 7 provide clear image please

12) Fig 8 is the final dosage reaching 100%

13) What is the significance of Table 9? Why R2 here? What logic it mentions?

14) Line 428 rewrite the sentence please

15) Line 433: Why are stability tests not taken?

Author Response

Reviewer #2:

1) What exactly authors mean by "asphalt mortar and mixture"? Mixture? or matrix?

Response: Thanks for your comment. Asphalt mortar is an asphalt-filler system formed by mixing asphalt and filler according to certain mass ratio. Besides, asphalt mixture is composed of asphalt, aggregate and filler.

2) Line 13: that are often: is often

Response: Thanks for your comment. According your suggestion, we have corrected the incorrect use of phrases in line 13.

3) As per l 13: it is already used widely in bitumen matrix, then what is the novelty of this work?

Response: Thanks for your comment. Although phosphogypsum is widely used in bitumen matrix, the global production of phosphogypsum exceeds 280 million tons per year, and the utilization rate of phosphogypsum in industry, agriculture, building materials and other industries does not exceed 15% of the output.

On the other hand, combine phosphogpsum and steel slag powder, It is positive for consuming phos-phogypsum and steel slag solid waste and beneficial for environmental protection.

4) How the mix composition is arrived? Is it based on DOE or literature or by preliminary tests?

Response: Thanks for your comment. The mix composition was designed in the preliminary test, corresponding statement had been added in the test.

5) Why phospogypsum is mixed with steel slag, what benefits authors expected, and what property of this two materials made authors decide this?

Response: Thanks for your comment. Phosphogypsum lead to poor moisture stability in asphalt material due to its acidity. However, Steel slag powder is a by-production of steel manufacture, it has alkalinity and good mechanical properties.Therefore, we considering use steel slag powder to neutralize acidic substances in phosphogypsum and to develop a phosphogypsum based filler. So, combine phosphogpsum and steel slag powder, It is not only positive for consuming phosphogypsum and steel slag solid waste , but also beneficial for environmental protection.

6) What is the importance of experiment done and explained via fig 4?

Response: Thanks for your comment. We believed that it is necessary to characterize effect of the phosphogypsum based filler (PF) on adhesion between asphalt and limestone. Since phosphogypsum was acidic and supposed to show poor moisture stability. How steel slag powder affected adhesion between PF based asphalt mortar and limestone can be hence acknowledged.

7) Line 235-237 is it relative density of soil formula? On what literature basis this calculation is done?

Response: Thanks for your comment. This is not relative density of soil formula. It is the calculation of overall desirability, a method that determine the optimal content of PF in replacing traditional limestone filler in consideration of physical property of asphalt mortar. The references 27 and 28 can be used to explain its calculation, and how it was used to calculate the optimal content of PF was already involved in the text.

8) Lot of articles content is about explaining the process of experiments, results and discussion is too limited, instead of this authors can simply mention the code used for the test unless the test is defined by authors or used newly which is not covered in a code.

Response: Thanks for your comment. We agree that redundant description of test can be omitted. On the other hand, discussion can be further enhanced. Corresponding revision had been done in the text.

9) Line 297: phogypsum’s high hydrophilicity may lead to poor adhesion between asphalt and aggregate: Then why is this material prescribed if it affects the bonding which in turn will affect strength?

Response: Thanks for your comment. The previous introduction did not clearly point out that the purpose of our study is to use phosphogypsum as material of filler. Therefore, the introduction had been greatly revised to highlight the meaning and purpose of this study. Phosphogypsum can be consumed and environment protection can be enhanced through its utilization in filler. How to alleviate its negative effect on moisture stability of asphalt material require further investigation, which hindered its application in road engineering. Therefore, this study tried to develop the phosphogypsum based filler that can meet the requirement for asphalt mixture.

10) Fig 6 interpreting is confusing.

Response: Thanks for your comment. Functional curve that fitting the data of penetration and softening point of P-AM was used to discuss effect of steel slag powder content intuitively, which was shown in Fig 6, namely Fig 4 in the revised manuscript. Data tendency can be hence simulated in the fitting curve and the optimal content of steel slag powder can also be found.

11) Fig 7 provide clear image please

Response: Thanks for your comment. A clearer image was put in the revised manuscript.

12) Fig 8 is the final dosage reaching 100%

Response: Thanks for your comment. Yes, the final dosage of steel slag powder was 100%, which means phosphogypsum had been completely replaced by steel slag powder.

13) What is the significance of Table 9? Why R2 here? What logic it mentions?

Response: Thanks for your comment. Table 9 includes the results of PI and T800. Both of them were calculated based on the linear equation fitting the penetration under different temperature. It was stipulated in the specification of JTG E20-2011 that the linear regression correlation coefficient R2 in equation (2) must not be less than 0.997. Therefore, it is necessary to confirm the determination coefficient R2 in Table 9 in order to confirm success of fitting.

14) Line 428 rewrite the sentence please

Response: Thanks for your comment. According to your suggestion, we reorganized the language and rewrote the sentence.

15) Line 433: Why are stability tests not taken?

Response: Thanks for your comment. T800 showed a very similar trend with PI. Therefore, only one parameter is needed to achieve the calculation, and repetition can be hence avoided.

Round 2

Reviewer 1 Report

The authors responded to reviewer suggestions and improved the manuscript.

The manuscript can be accepted for publication after shortening/ revising the Conclusion part. 

Author Response

Thanks for your constructive comments. According to your comments, we have revised the conclusion in detail and deleted the redundant parts.
